# WHY DOES THE VQA MODEL ANSWER NO?: IMPROVING REASONING THROUGH VISUAL AND LINGUISTIC INFERENCE

## ABSTRACT

In order to make Visual Question Answering (VQA) explainable, previous studies not only visualize the attended region of a VQA model but also generate textual explanations for its answers. However, when the model's answer is 'no,' existing methods have difficulty in revealing detailed arguments that lead to that answer. In addition, previous methods are insufficient to provide logical bases, when the question requires common sense to answer. In this paper, we propose a novel textual explanation method to overcome the aforementioned limitations. First, we extract keywords that are essential to infer an answer from a question. Second, for a pre-trained explanation generator, we utilize a novel Variable-Constrained Beam Search (VCBS) algorithm to generate phrases that best describes the relationship between keywords in images. Then, we complete an explanation by feeding the phrase to the generator. Furthermore, if the answer to the question is "yes" or "no," we apply Natural Langauge Inference (NLI) to identify whether contents of the question can be inferred from the explanation using common sense. Our user study, conducted in Amazon Mechanical Turk (MTurk), shows that our proposed method generates more reliable explanations compared to the previous methods. Moreover, by modifying the VQA model's answer through the output of the NLI model, we show that VQA performance increases by 1.1% from the original model.

## 1 INTRODUCTION

Recently, the interpretability and reliability of Artificial Intelligence (AI) are being seriously discussed (Kim et al., 2018b), since the opacity of its internal structure makes it hard to figure out the exact reason when it malfunctions (Gunning, 2017; Kurakin et al., 2016). Previous works show that decision process of conventional machine learning models (e.g., decision tree) can be modified to be transparent while maintaining their original performance (Si & Zhu, 2013; Letham et al., 2015). For deep learning models, Ribeiro et al. (2016) suggest a method for reasoning their decisions by visualizing critical parts of the input for the image classification. However, it is harder to infer the decision process for more complicated tasks such as VQA.

VQA is one of the most active research fields in AI with strong expectation for real applications. It can be used for either assisting radiologists (Letham et al., 2015) or help visually impaired people to understand their environment more flexibly (Antol et al., 2015). Therefore, state-of-the-art VQA models have been continually proposed, but their internal processes are yet opaque. Displaying attended region might be used for indirect reasoning. However, it has limitation on describing what the model perceives in detail. The most spotlighted recent research for endowing interpretability to VQA model is making the model to produce an answer and its explanation at the same time (Park et al., 2018; Li et al., 2018; Hudson & Manning, 2019). To make it possible, they provide a ground truth explanation dataset for each image, question and answer pairs, and fine-tune the model to generate explanations in an end-to-end process.

Previous explanation models can successfully describe relationships between objects in the image. However, when the answer to the question is "no," the previous methods have difficulty on providing logical bases. Also, even though there are questions that require common sense, existing models

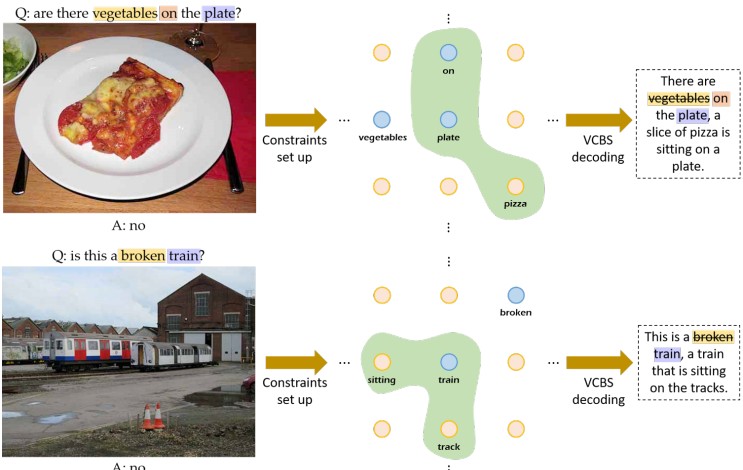

Figure 1: Sample results using the proposed method. We first identify several constraints (blue nodes) from the question and answer. Then, the proposed Variable-Constrained Beam Search (VCBS) algorithm searches the best possible sentence which satisfies the given constraints as far as possible.

do not utilize it: previous models memorize explanations appropriate to each situation in training datasets (Park et al., 2018; Zellers et al., 2019).

In this paper, we propose a novel method that generates explanations for a VQA model's answer. First, the proposed method selects keywords that play an essential role in inferring the answer to the question. We extract these keywords in the same way as Wang et al. (2019). Next, given a pre-trained explanation generator, we use the proposed Variable-Constrained Beam Search (VCBS) algorithm to generate a phrase that best describes the relationship between the keywords by inferring the image (See Fig. 1). The phrase is fed to the generator to complete the remaining part of the explanation. Finally, we use Natural Language Inference (NLI) (Wang et al., 2019; Talman et al., 2019) to check whether the contents of the question can be inferred from the explanation using common sense. For the overview of our method, please refer to Fig. 4.

Our main contributions are:

- We propose the novel method to generate textual explanations for the VQA model's answer. The proposed method not only produces more appropriate explanations (Kim et al., 2018b) but can also be used as a post-processing of the VQA model to boost the performance. We validate our method on one hundred randomly sampled data from VQA 2.0 test-std split, through a user study conducted in MTurk. By modifying the predicted answer using the generated explanation, we also compare the prediction accuracy of the method on complete VQA 2.0 test-std split. Our proposed method has gained a MOS score higher than previous methods, and get a VQA performance improvement of about 1.1%.

- We propose the Variable-Constrained Beam Search (VCBS) algorithm that helps to identify the visual relationship between objects and attributes of interest in the image. Unlike the previous Constrained Beam Search (CBS) algorithm (Sipser et al., 2006), the VCBS has the *variable accepting states*. The variable accepting states help to satisfy the given constraint as much as possible but allow the final beam to exclude some constraints if necessary.

## 2 RELATED WORKS

We first summarize recent research in VQA. Then we review the explanatory model for VQA and discuss the distinguishing characteristics of our work.

**Visual Question Answering:** The VQA challenge was first launched in 2016, and representative studies at that time (Fukui et al., 2016; Lu et al., 2016; Kim et al., 2017) first extract image features

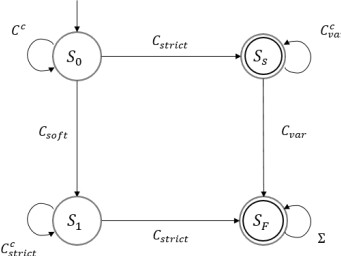

Figure 2: The state diagram of the proposed VCBS algorithm: the $\Sigma$ is the complete vocabulary, $C_{strict}$ is a set of strict constraints, $C_{soft}$ is a set of soft constraints, $C = C_{strict} \cup C_{soft}$ is the whole constraints. $C_{var} = \{w_t | | (\{w_0, \cdots, w_{t-1}\}, w_t) \cap C | \geq |C| - p\}$ is a set of the variable constraints, where $(\{w_0, \cdots, w_{t-1}\}, w_t) = \{w_0, \cdots, w_t\}$, $w_t$ is the next output word from the captioner, and $p$ is the constraint loosening parameter. Note that $S_S$ and $S_F$ are set of states (See Fig. 4). Only the beams in accepting states in $S_F$ can used as the feeding phrase (Sipser et al., 2006; Anderson et al., 2018a; 2017).

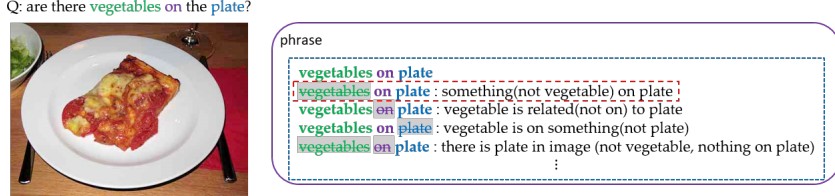

Figure 3: Our method can specify the logical basis for the question when the answer is "no."

using residual networks (He et al., 2016) and established joint learning through question features and attention obtained from gated recurrent units (GRUs). In the VQA 2017 challenge, Anderson et al. (2018b) proposed a method that exploits Faster-RCNN (Ren et al., 2015) to extract image features for the attended object candidates. Following its scheme, most of the subsequent VQA studies also used Faster-RCNN. In the VQA 2018 challenge, Kim et al. (2018a) presented bi-linear attention so that both question features and visual features were considered together when generating attention. They have shown that bilinear attention improves the residuality of the attention presented in (Kim et al., 2016) successfully.

Although various attention-based VQA models have shown excellent performance, the models heavily rely on linguistic bias rather than visual information in most cases (Ramakrishnan et al., 2018). For this reason, recent studies are interested in identifying relationships among objects in images and using them to predict answers (Hudson & Manning, 2019).

**Explanatory model for VQA:** Visualizing attention (Lu et al., 2016) or utilizing Grad-CAM (Selvaraju et al., 2017) is one of the most widespread approach to understand the decision process of the VQA model. However, as mentioned in Li et al. (2018), we need detailed textual explanations rather than merely visualizing these areas. For this reason, recent research not only released ground truth explanation datasets (VQA-X (Park et al., 2018) and VQA-E (Li et al., 2018)) but also proposed models that generates an answer and its textual explanation at the same time. While VQA-X is made up of more elaborate and detailed explanations, VQA-E is composed of more extensive training and validation data. According to the results of the Li et al. (2018), fine-tuning through the VQA-E dataset increases the prediction accuracies of VQA models noticeably.

While both of the models are good at justifying the answer for the given question and image, they are poor at providing introspective explanations of the VQA model. First, the PJ-X model proposed by Park et al. (2018) consists of the sequence of a VQA model and an explanatory model. Since the PJ-X does not refer to the attention generated in the VQA model, the explanation does not reflect the decision process of the target VQA model. On the other hand, the multi-task VQA-E model proposed by Li et al. (2018) directly utilize the attention distribution of pre-trained VQA models. Although the model generates explanations based on visual information about the attended area, it

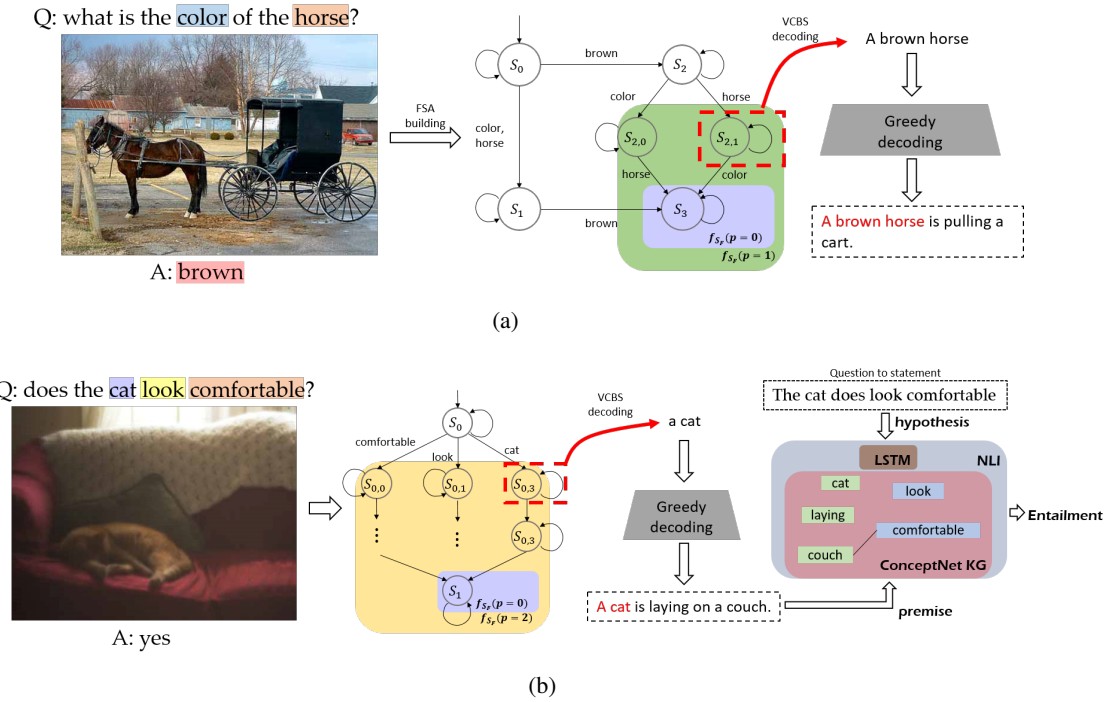

Figure 4: Overview of the proposed method. (a) If the answer is not yes/no, we set the answer as a strict constraint in VCBS decoding. (b) Else, we do not use a strict constraint. Note that we use linguistic inference only if the answer is yes/no.

often produces sentences that are not related to the question. This is because multi-task VQA-E does not directly specify the relationship between the objects related to the question.

Since it is important for VQA task to grasp the relationship between specific objects and attributes in an image, recent studies infer an answer through a scene graph or a modular network formed for the image (Hudson & Manning, 2019; Hu et al., 2018). Although both of them provides explanatory insights into the model, the methods lack a logical basis for the explanation in situations where the answer is "no." (See Fig. 3).

**Natural Language Inference:** The goal of NLI is to classify the relationship between two sentences: "premise" and "hypothesis." If the "hypothesis" can be inferred from the "premise," we classify the relationship between the two sentences as "entailment." While most NLI models use LSTM to classify the relationship between the two sentences (Talman et al., 2019), Wang et al. (2019) proposed a method to grasp the relation by using ConceptNet (Speer et al., 2017) (a common sense knowledge graph obtained through crowd-sourcing) as external knowledge. We use state-of-art NLI models (the HBMP (Talman et al., 2019), and the ConSeqNet (Wang et al., 2019)) to identify if the question could be answered using common sense and our explanation (See Fig. 4(b)).

## 3    APPROACH

For an input image, the Recurrent Neural Network (RNN) in the captioner forms the Variable Order Markov (VOM) model between words (Begleiter et al., 2004). This VOM model can be represented as a directed graph $G = (\Sigma, E)$, where $\Sigma$ is the complete vocabulary, and $E$ is probability distribution at each node (See Fig. 1). If the captioner is ideal, the goal of VQA can be regarded as (i) checking whether a path that contains specific nodes can be found (for the question with yes/no answers. See Fig. 1) ; (ii) or completing the path that pass through the specific nodes (for the other type of questions. (Lin et al., 2019)).

---

**Algorithm 1** Variable-Constrained Beam Search decoding

---

1: **procedure** VCBS$(\Theta, b, T, C, A = (\sum, S, s_0, f_{S_F}(p), \delta))$
2:   $M \leftarrow \phi$                    ▷ for Memoization
3:   **for** $s \in S$ **do**
4:    $B^s \leftarrow \{\epsilon\}$ if $s = s_0$ else $\phi$
5:    **for** $t \leftarrow 1$ to $T$ **do**
6:     $E^s \leftarrow \bigcup_{s' \in S}\{(y', w)|y' \in B^{s'}, w \in \sum, \delta(s', w) = s\}$
7:     $B^s \leftarrow \arg\max_{E' \subset E^S, |E'|=b} \sum_{y^{E'} \in E'} \Theta(y^{E'})$     ▷ Beam extension
8:     **if** $s \in f_{S_F}(p)$ **then**       ▷ for Beams in accepting states
9:      $M \leftarrow \arg\max_{y^{M \cup B^s} \in M \cup B^s} |C \cap y^{M \cup B^s}|$   ▷ keep beams satisfying more constraints
10:      $M \leftarrow \arg\max_{M' \subset M, |M'|=1} \sum_{y^{M'} \in M'} \Theta(y^{M'})$   ▷ only keeps the best beam
11:   **return** $y \in M$ if $M \neq \phi$ else $\phi$

---

## 3.1 CONSTRAINTS

First, we identify Part-Of-Speech (POS) tags (Màrquez & Rodríguez, 1998) of each word in the question. Then we take all detected nouns and verbs (excluding auxiliary verbs) as keywords. If the VQA model's answer is "yes" or "no," we add adposition to keywords (See Fig. 1). Otherwise, we use the answer as additional keywords (See Fig. 4(a)).

As in the work of Anderson et al. (2018a), we construct Wordnet synsets (Fellbaum, 2010) containing plural and singular forms for each keyword. Thus, the nodes in Fig. 1 correspond to the synsets of each keyword. We use a set of these synsets as constraints set $C$ in our VCBS algorithm. For simplicity in notation, we assume that $C$ is composed of keywords, not synsets, in the rest of this paper. The set $C$ is further divided into two subsets, strict constraints ($C_{strict}$) and soft constraints ($C_{soft}$). $C_{strict}$ is a set of keywords that the final beam must satisfy in order to be in accepting state (See Fig. 2). $C_{soft}$ is a set of keywords that are recommended to satisfy in order for the final beam to be accepted, but not necessarily required. We use the answer as strict constraints only if the answer is not "yes" or "no" (See Fig. 4).

## 3.2 VARIABLE-CONSTRAINED BEAM SEARCHING

For the RNN in image captioner with parameter $\theta$, a word $w_t$ at time step $t$ is predicted given the previous sequence of words $\{w_0, \cdots, w_{t-1}\}$ and the image $I$.

$$w_t = \arg\max_{\hat{w}_t} p_\theta\left(\hat{w}_t \mid w_{t-1}, \cdots, w_0, I\right), \tag{1}$$

To naturally place the keywords into the sequence, we need to partially specify sequence with Beam Search decoding (Anderson et al., 2018a). During the completion of a beam, whenever the beam satisfies a constraint, we change the state of the beam (See Fig. 4). We denote this state-transition function as $\delta$. The completed beam for a given image may not satisfy all constraints: no clue is found in the image that satisfies the constraint (See Fig. 1.), or the constraint is given as a hyponym (See Fig. 4(a)). Therefore, the satisfaction of constraints for each beam is not compulsory, but recommended (except the $C_{strict}$).

We express this condition as a set of accepting states $S_F = f_{S_F}(p)$ with a variable $p$, where $f_{S_F}(p)$ : $\mathbb{N} \rightarrow 2^S$, and $S$ is a set of all possible states. The loosening parameter $p$ denotes the number of keywords in $C$ that can be neglected. The overview of our VCBS decoding is depicted in Algorithm 1. Note that $\Theta = log\, p_\theta(\cdot)$, $b$ is the beam size, $T$ is the maximum sequence length. Since the completed beams may not stem from the phrase that satisfies the constraint, we keep the best phrase presents in $S_F$ through Memoization. Finally, we complete the phrase $y \in M$ through the greedy decoding in Equation 1.

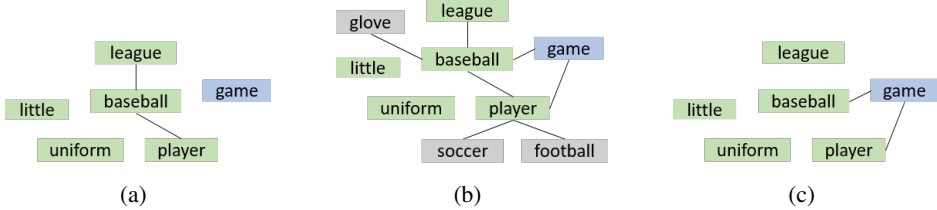

Figure 5: An example illustration of subgraphs in Fig. 6(c): (a) Concepts Only, (b) One-Hop, (c) One-Hop Direct. Green nodes are concepts in the "premise," and the blue nodes are concepts in the "hypothesis." Gray nodes are concepts in the ConceptNet KG (Speer et al., 2017) that do not belongs in both cases.

## 3.3 LINGUISTIC INFERENCE

Some of the questions in VQA require inferences based on background knowledge in addition to visual information. Most of these types of questions require complex inference processes (Lin et al., 2019) but can be reasoned relatively quickly when the answer is yes/no : by checking whether the contents of the question can be inferred from visual information. Consequently, we first convert the question to a statement by moving the auxiliary verb to behind the subjective noun (e.g., Are dogs on the grass? ⇒ Dogs are on the grass). Using this statement as "hypothesis" and the complete sentence from $y$ as "premise," we checked whether they are in the "entailment" relationship (See Fig. 6(c)).

For the knowledge extension, we extract concepts from "premise" and "hypothesis," and identified their relationship in ConcepNet Knowledge Graph (ConceptNet KG). As in Wang et al. (2019), we use nouns and verbs (excluding auxiliary verbs) as concepts for "premise." However, for "hypothesis," we only use elements of remaining constraints $C \cap y^c$ as concepts (See Fig. 4(b)). Finally, we use Concepts Only (Wang et al., 2019) and *One-Hop Direct* subgraphs with knowledge graph embedding (Wang et al., 2014) for NLI. The One-Hop Direct subgraph only connects concepts from "premise" to "hypothesis" in one hop (See Fig. 5). These approaches reduce trivial subject-predicate-object (SPO) triples (Lu et al., 2019) significantly, hence, increase the performance of the NLI model for VQA.

## 4 EXPERIMENTS

### 4.1 DATASETS AND IMPLEMENTATION

**VQA model:** We use two-glimpse BAN (Kim et al., 2018a) for the target VQA model. All hyperparameter settings follow the previous work (Kim et al., 2018a). The input image features are extracted from the Faster-RCNN (Ren et al., 2015), pre-trained using Visual Genome dataset (Krishna et al., 2017). We adaptively extract 10 to 100 objects per image, as in (Kim et al., 2018a). For the question embedding in VQA, we initialize 300 dimension word embeddings with pre-trained GloVe vectors (Pennington et al., 2014).

**Explanation Generator:** We use the Up-Down captioning model (Anderson et al., 2018b) as our explanation generator. For training the model, we use both train and validation splits of MS COCO captioning dataset (Chen et al., 2015). While building the vocabulary, we drop any words if they occur less than five times in the training dataset, resulting in 10,512 words in $\sum$.

**VCBS decoding:** Since all questions have a different number of constraints $|C|$, we repeat the VCBS decoding with increasing values of $b$ and $p$ until $M$ is not empty or $p = |C|$: we initiate with $b = 3$ and $p = 0$ (i.e., $(b, p) = \{(3, 0), (4, 1), (5, 2), (6, 3), (6, 4), \cdots, (6, |C|)\}$). Note that we set an upper limit of $b$ as 6: beam size larger than 6 decreases the decoding speed drastically.

**NLI model:** We train both the HBMP (Talman et al., 2019) and the ConSeqNet (Wang et al., 2019) on the combination of multiple datasets: SNLI dataset (Bowman et al., 2015), SciTail dataset (Khot et al., 2018), and MultiNLI dataset (Williams et al., 2018). While generating explanations, we only

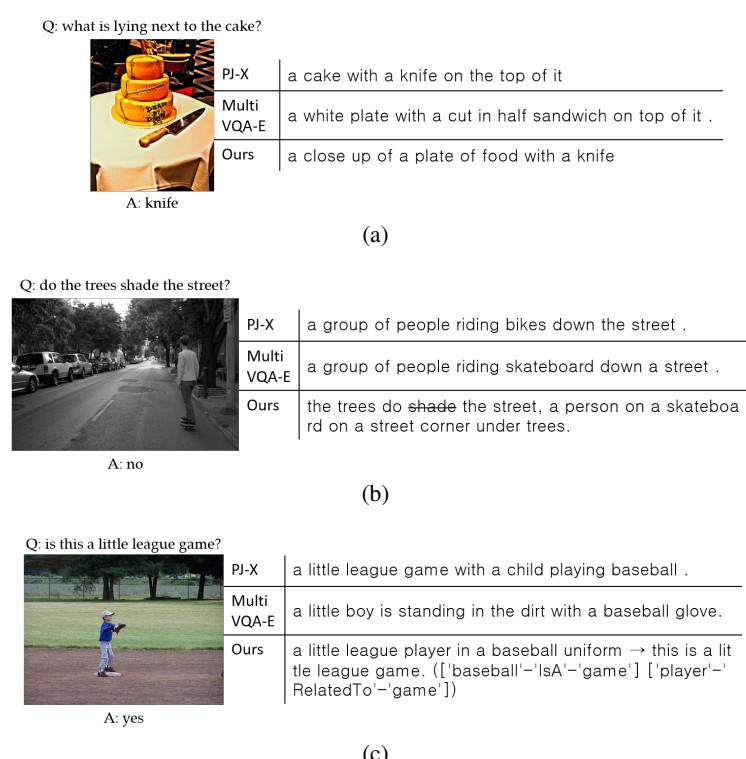

Figure 6: Sample examples of VQA explanations using PJ-X (Park et al., 2018), Multi-task VQA-E (Multi VQA-E) (Li et al., 2018), and the proposed method. We compare explanations for three cases: (a) for the usual questions and answers, (b) when the answer to the question is "no," (c) when the answer requires linguistic inference. If the "premise" and "hypothesis" are in an "entailment" relationship, we denote it with "premise" → "hypothesis." We also visualize the SPO triples (Lu et al., 2019) in the One-Hop Direct subgraph (i.e., ['subject'-'predicate'-'object']) to improve the explainability of the proposed method.

use the ConSeqNet (Wang et al., 2019) to visualize the linguistic inference process by One-Hop Direct subgraph (See Fig. 6(c)).

**Evaluation of VQA Explanations:** Park et al. (2018) and Li et al. (2018) evaluated the performance by measuring the similarity between the generated description and its ground truth, using ordinary metrics in image captioning (i.e., CIDEr (Vedantam et al., 2015)). Measuring the CIDEr score for VQA-X or VQA-E is a measure of how well the justified explanation is reproduced. Since we focus on generating explanations through visual and linguistic inference, we use the Mean Opinion Score (MOS) to compare with these methods on one hundred random samples from VQA 2.0 test-std split (He et al., 2016; Antol et al., 2015).

To minimize the bias in the survey, we conduct an online user study in MTurk (Paolacci et al., 2010), using a hundred random samples from VQA 2.0 test-std split. According to our preliminary survey within our research group, survey participants were likely to lost focus on the study if they were asked a hundred questions without a rest. For this reason, we divide 100 questions into five splits (i.e., 20 questions for each split). The study involved 52 Amazon Mechanical Turk workers (Turkers) for each split, who spent an average time of about 12 minutes to complete a split. For each explanation, we ask its score for suitability as reasoning, ranging from 1 to 5 (i.e., 1 indicates a very inappropriate explanation, and 5 corresponds to a very appropriate explanation).

## 4.2 QUALITATIVE RESULTS

We compare the performance of the proposed method with state-of-the-art methods (PJ-X (Park et al., 2018), Multi-task VQA-E (Multi VQA-E) (Li et al., 2018)), and visualize the difference

between the previous methods and the proposed method in Fig. 6. When the answer to the question is "no," previous methods only describe the image, but our method clarifies the reason (i.e., the model cannot find shade from the image) (See Fig. 6(b)). Figure 6(c) shows an example when the output of the NLI model is "entailment." The proposed method first creates a description of the objects and attributes of interest in the image. Then it shows how the answer to the question is inferred from the description.

## 4.3 QUANTITATIVE RESULTS

Table 1: Mean opinion score (MOS) comparison on one hundred random samples from VQA 2.0 test-std split.

| Method | split 1 | split 2 | split 3 | split 4 | split 5 | overall |
|---|---|---|---|---|---|---|
| PJ-X (Park et al., 2018) | 2.941 | 3.2749 | 2.9987 | 2.9410 | 2.6095 | 2.9529 |
| Multi VQA-E (Li et al., 2018) | 2.9038 | 3.0625 | 2.794 | 2.9108 | 2.5125 | 2.8367 |
| Ours+ConSeqNet (Wang et al., 2019) | **3.5194** | **3.4532** | **3.8415** | **3.4992** | **3.691** | **3.6089** |

Table 2: MOS score categorization by answer types.

| Model | Yes/no | Number | Other |
|---|---|---|---|
| PJ-X (Park et al., 2018) | 2.5154 | 3.3476 | **3.5145** |
| Multi VQA-E (Li et al., 2018) | 2.4983 | **3.3754** | 3.4189 |
| Ours+ConSeqNet (Wang et al., 2019) | **4.1321** | 3.3571 | 3.3909 |

Table 3: Test-dev scores of models on VQA 2.0 dataset. We use two-glimpse BAN (Kim et al., 2018a) for the experiment. Note that the proposed methods use more data in training (i.e., SNLI dataset (Bowman et al., 2015), etc), and are not fairly compared with the BAN.

| Model | Overall | Yes/no | Number | Other |
|---|---|---|---|---|
| BAN (Kim et al., 2018a) | 68.13 | 84.44 | 53.38 | 57.50 |
| Ours+HBMP (Talman et al., 2019) | 65.48 | 78.04 | 53.38 | 57.50 |
| Ours+ConSeqNet (Wang et al., 2019) | **69.23** | **87.10** | 53.38 | 57.50 |

We show the MOS results in Table 1. The proposed method attained the highest score, while the PJ-X model (Park et al., 2018) attained similar score as the Multi VQA-E model (Li et al., 2018). We also categorize the MOS scores according to the answer types in Table 2. The result shows that our method for questions of yes/no answers has distinct advantages over the previous methods.

Finally, if the VQA model's answer is yes/no, we change the answer to "yes" if the output of the NLI model is "entailment" and to "no" otherwise (if $p = |C|$, we do not change the answer). Table 3 shows the VQA performance comparison of models on the test server submission. The VQA performance is reduced when using the HBMP (Talman et al., 2019) for NLI, but the VQA performance is increased when the ConSeqNet (Wang et al., 2019) is used for NLI. Since the ConSeqNet is trained using broader background knowledge (the HBMP does not use the ConceptNet KG (Speer et al., 2017)), this result shows that background knowledge is essential for answering yes/no in VQA.

## 5 CONCLUSION

In this paper, we have suggested a novel method that generates explanations for VQA using visual and linguistic inference. Firstly, we describe the relationship between objects and attributes of interest in the image using the proposed VCBS decoding. The VCBS algorithm helps to generate explanations that contain essential concepts from the question. Next, if the answer is yes/no, we use NLI to check whether the content from questions can be inferred from the explanations. If it can be inferred, we visualize a One-Hop Direct subgraph between the explanation and the question. We evaluate the performance of the proposed method by measuring MOS. For the fair comparison, we conduct an online user study using MTurk. The result shows that our model attains better performance than the previous models. Finally, we modified the VQA model's answer based on the

NLI output and measured the performance. For the state-of-art NLI model (Wang et al., 2019) using ConceptNet KG (Speer et al., 2017), we show an increase in the VQA model's performance.

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

# A APPENDIX

## A.1 QUANTITATIVE RESULTS

We used the BUTD model trained on COCO captioning data (Chen et al., 2015) as an explanation generator. The reasons for using it are as follows: 1) The COCO dataset is much larger than the VQA-X (Park et al., 2018) and VQA-E (Li et al., 2018), so we believe that the model trained on the COCO dataset can better understand the relationship between objects in an image. 2) We cannot train the Multi-VQAE (Li et al., 2018) (or PJ-X (Park et al., 2018)) model on the COCO captioning dataset, because the dataset does not include VQA answers. However, we conduct additional experiments to confirm that our method can be used to improve the performance of the existing textual explanation model.

For this purpose, we extract 100 additional experimental samples from the VQA 2.0 test-std split. These samples are chosen so as not to duplicate the samples used in previous experiments. Similar to the experiment in Table 1, we divided 100 samples into five splits. Unlike before, this survey involved 40 AMT workers for each split. Note that we used two different explanation generators in this experiment: the Up-Down (Anderson et al., 2018b) model trained on COCO captioning dataset (Chen et al., 2015), and the Multi-VQAE (Li et al., 2018) trained on VQA-E dataset (Li et al., 2018). The results in Table 4 and Table 5 show that our method helps to generate more appropriate textual explanations, especially when the VQA model's answer is "yes" or "no."

## A.2 QUALITATIVE RESULTS

We show qualitative results from the experiment conducted in Table 4 (See Fig. 7 and Fig. 8).

Table 4: Mean opinion score (MOS) comparison on one hundred random samples from VQA 2.0 test-std split. Note that we used two different explanation generators: the Up-Down (Anderson et al., 2018b), and the Multi-VQAE (Li et al., 2018).

| Method | split 1 | split 2 | split 3 | split 4 | split 5 | overall |
|---|---|---|---|---|---|---|
| Multi VQA-E (Li et al., 2018) | 2.9589 | 3.2089 | 3.1339 | 3.0835 | 3.0292 | 3.0829 |
| Ours (Explanation Generator : Up-Down)+ConSeqNet | **3.2284** | **3.4495** | **3.4712** | **3.3818** | **3.2932** | **3.3648** |
| Ours (Explanation Generator : Multi VQA-E)+ConSeqNet | 3.0146 | 3.3130 | 3.1450 | 3.1090 | 2.9795 | 3.1122 |

Table 5: MOS score (in Table 4) categorization by answer types.

| Model | Yes/no | Number | Other |
|---|---|---|---|
| Multi VQA-E (Li et al., 2018) | 2.9875 | 2.8625 | 3.1951 |
| Ours (Explanation Generator : Up-Down)+ConSeqNet | **3.3511** | **2.9990** | **3.4304** |
| Ours (Explanation Generator : Multi VQA-E)+ConSeqNet | 3.0465 | 2.8544 | 3.2122 |

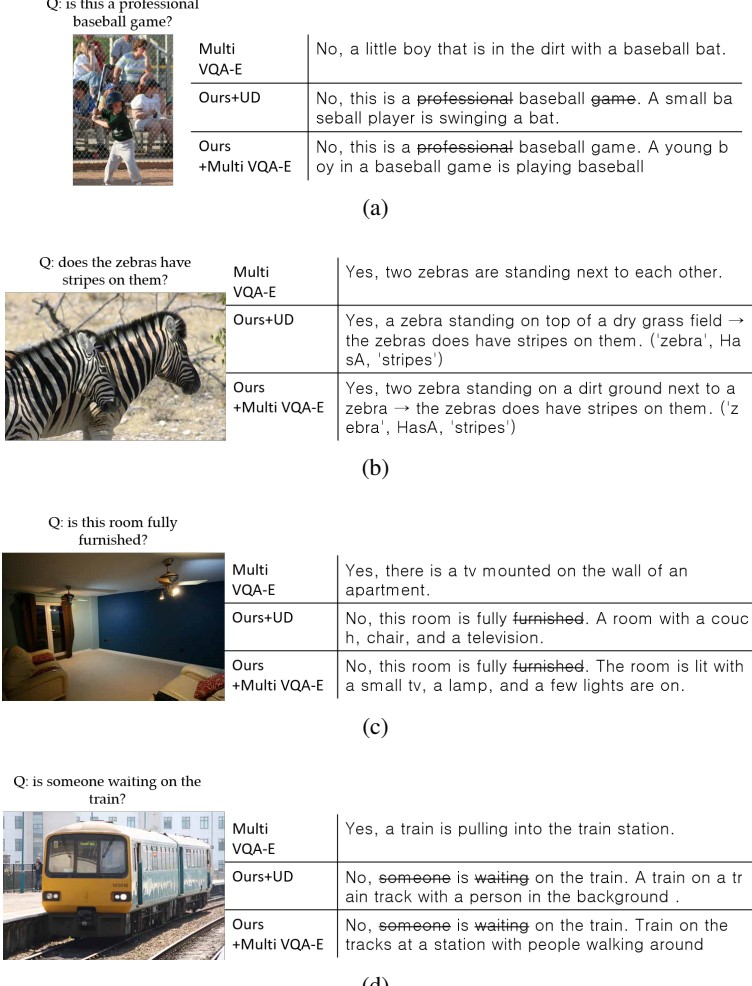

Figure 7: Sample examples of VQA explanations using Multi-task VQA-E (Multi VQA-E) (Li et al., 2018), our method using the Up-Down model (Anderson et al., 2018b) as the explanation generator, and our method using Multi VQA-E as the explanation generator.

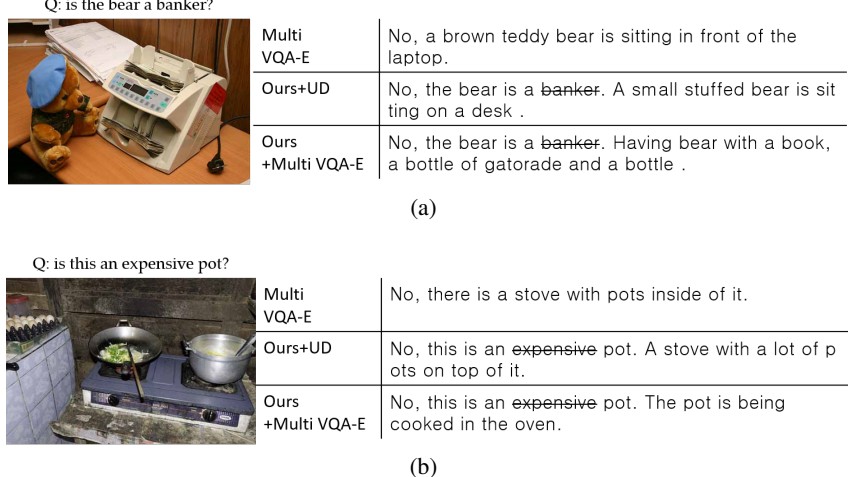

(a)

(b)

Figure 8: Sample examples of VQA explanations using Multi-task VQA-E (Multi VQA-E) (Li et al., 2018), our method using the Up-Down model (Anderson et al., 2018b) as the explanation generator, and our method using Multi VQA-E as the explanation generator. It is shown that the Up-Down model trained on COCO captioning dataset better understands relationships between objects in the image.

### A.3 STRICT ENTAILMENT AND WEAK ENTAILMENT

We conducted additional experiments with using the VQA model's answer as "yes," if the class probability of the NLI model for "Entail" is greater than "Contradictory," even though the NLI result is "Neutral" (Weak Entailment). In this experiment, we found that the answer is often "yes," even when there is little correlation between premise and hypothesis. For this reason, we use the answer as "yes" only if the NLI result is "Entail" (Strict Entailment) (See Fig. 9).

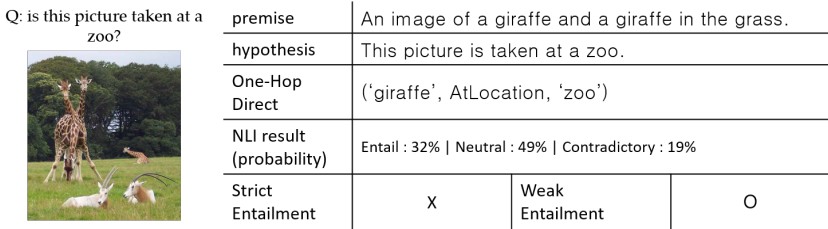

Figure 9: Comparison between strict entailment and weak entailment. In this paper, we regard the premise and hypothesis are in the "entailment" relationship only if the NLI result is "Entail."

