# OpenReview forum: "Why Does the VQA Model Answer No?: Improving Reasoning through Visual and Linguistic Inference"
_ICLR.cc/2020/Conference — Reject_

### Official Review · AnonReviewer2 · 2019-10-22
**Official Blind Review #2**

**Rating:** 6

**Review:**


The paper proposes a novel method for explaining VQA systems. Different from most previous work, the proposed approach generates textual explanations based on three steps. First, it extracts keywords from the question. Then an explanation sentence is decoded (based on an RNN image captioner) through the proposed Variable-Constrained Beam Search (VCBS) algorithm to satisfy the keyword constraints. Finally, 3) checking through linguistic inference whether the explanation sentence can be used as a premise to infer the question and answer.

I would recommend for acceptance. The paper proposes an alternative approach to VQA explanations, together with a few supporting algorithms such as VCBS. It is potentially helpful to future work on textual explanations and explainable AI in general.

At a high level, it is ambiguous to decide what is a reasonable explanation for many “no” answers. For example, one usually cannot provide stronger justification than “there is indeed no one” or “I don’t see anyone” to the question “Is there anyone in the room” with an answer “no.” The paper frames this explanation generation task as a linguistic inference task and checks entailment between the explanation and the question-answer pair. While it is debatable whether this is optimal, the proposed approach provides valuable insights on what constitutes a good explanation.

However, the proposed approach also has noticeable weaknesses.

It relies on external models or tools for natural language inference, and such inference does not take into account the visual context of the image. Also, the explanations generated from the proposed model only justify the answer but are not introspective, and they do not reflect the decision process of the target VQA model.

**Experience Assessment:**

I have published one or two papers in this area.

**Review Assessment: Checking Correctness Of Derivations And Theory:**

N/A

**Review Assessment: Checking Correctness Of Experiments:**

I carefully checked the experiments.

**Review Assessment: Thoroughness In Paper Reading:**

I read the paper at least twice and used my best judgement in assessing the paper.

---

> ### Author Response · Authors · 2019-11-14
> **Response to Review #2**
>
> We are hugely thankful for your invaluable and thoughtful comments and support.
>
> 1) However, the proposed approach also has noticeable weaknesses. It relies on external models or tools for natural language inference, and such inference does not take into account the visual context of the image. Also, the explanations generated from the proposed model only justify the answer but are not introspective, and they do not reflect the decision process of the target VQA model.
>
> We agree with the issue the reviewer has pointed out. It's best to do inference using both visual and linguistic information at the same time, but unfortunately, we have not solved this problem yet. Therefore, we proposed a method of inference by using Visual and Linguistic information step by step. We also agree that explanations created in the proposed way are not introspective.  For this reason, we have revised the content of this paper as follows: (the proposed method not only produces more introspective explanations, ... ->  the proposed method not only produces more appropriate explanations, ...)

---

### Official Review · AnonReviewer3 · 2019-10-23
**Official Blind Review #3**

**Rating:** 6

**Review:**


 I thank the authors for their response.  I would keep my score unchanged (i.e., 6 Weak Accept).

-----------------------------------------------

Strengths:
- The paper enhances the beam search approach to generate explanations for answers to visual questions. The explanations are further used for verifying the yes/no answers.
- The paper constructs a VCBS algorithm with novelties in allowing soft constrains of the generated beams. Since I have not worked on the constrained beam search before, thus it is hard for me to measure the novelty of this method.
- The results of VQA 2.0 is pretty good. The accuracy of the Yes/No questions almost achieves the SotA systems.


Weakness:
- I have a question about the NLI system. Since the NLI is a three-way classifier where the answers would be "Entail", "Contradictory", and "Neutral".  What would the system do when the relationship is "Neutral"? For now, I think that it just give an answer of "no" but I am not sure whether it is correct.  For example, in Fig. 3, it shows an explanation (premise) of "something (not the vegetable) on a plate" and the hypo is "there are vegetables on the plate". Since the hypo is not necessary to contradict the premise, the relationship should be neutral. It does not directly provide evidence of the answer.

Comments:
- Since more data are involved in training the explanation system, the proposed methods are not fairly compared with the BAN method. It would be better to mention this detail in the paper.



**Experience Assessment:**

I have published one or two papers in this area.

**Review Assessment: Checking Correctness Of Derivations And Theory:**

I assessed the sensibility of the derivations and theory.

**Review Assessment: Checking Correctness Of Experiments:**

I assessed the sensibility of the experiments.

**Review Assessment: Thoroughness In Paper Reading:**

I read the paper at least twice and used my best judgement in assessing the paper.

---

> ### Author Response · Authors · 2019-11-14
> **Response to Review #3**
>
> We are obliged to get your suggestion and advice, and it is an honor to have your assistance.
>
> 1) I have a question about the NLI system. Since the NLI is a three-way classifier where the answers would be "Entail", "Contradictory", and "Neutral". What would the system do when the relationship is "Neutral"? For now, I think that it just give an answer of "no" but I am not sure whether it is correct. For example, in Fig. 3, it shows an explanation (premise) of "something (not the vegetable) on a plate" and the hypo is "there are vegetables on the plate". Since the hypo is not necessary to contradict the premise, the relationship should be neutral. It does not directly provide evidence of the answer.
>
>
> Following the reviewer's thoughtful concern, we conducted additional experiments to confirm this. In this experiment, we set the VQA model’s answer as "yes" if the class probability of the NLI model for "Entail" is greater than "Contradictory," even though the NLI result is "Neutral."
>
> As the reviewer pointed out, we found that the answer is often "yes," even when there is little correlation between premise and hypothesis. For this reason, we use the answer as "yes" only if the NLI result is "Entail,” as we first suggested in the paper.
>
> p.s-
> In Fig 3, the premise is "a slice of pizza is sitting on a plate." "something (not the vegetable) on a plate" is a sentence added to visualize which keyword is absent in the image.
>
> 2) Since more data are involved in training the explanation system, the proposed methods are not fairly compared with the BAN method. It would be better to mention this detail in the paper.
>
> Thank you for your advice. We mentioned the reviewer's comment in Table 3.

---

### Official Review · AnonReviewer1 · 2019-10-24
**Official Blind Review #1**

**Rating:** 3

**Review:**

Interesting results and analysis but lack of novelty and details.

This paper proposed a novel text explanation method which extracts keywords that are essential to infer an answer from question. The authors proposed VCBS based on CBS and use Natural language inference to identify the entailments when the answer is yes or no. Experiment results show better mean opinion score 100 random samples results compared with the previous method and better VQA binary classifications when flipping based on the NLI results.

Different from [Park et. al. 2018], who collected the explanations, this paper directly use the coco captions dataset as the source for the explanations. One of my major concern about this paper is it actually generate relevant captions with respect to the question and answer instead of real explanations. It's true that correlated caption can sometimes serve as the explanation when they cover the same concept coincidently but, we can not use these captions to explain the reasoning process of VQA.

The technique novelty of the proposed paper is also limited, the major novelty is the VCBS, which seems very similar to CBS. The only difference is VCBS adds relaxed parameters, which seems no technique novelty. Most annotations in Algorithms 1 is also not explained, making the readers hard to follow the actual content. The proposed model, although tied the vqa words with the explanation words, it suffers the same problems as PJ-X model, which didn't consider the VQA attention at all.

The experiment is also weak, considering the results is conduct on 100 samples, there might be significant variance. It's interesting the compared approach is learned based on a different dataset, which makes the results harder to compare. The NLI model results are interesting but for a more fair comparison, I would expect the proposed method compare with a model trained with VQA and coco caption dataset, such as VQA-E.



**Experience Assessment:**

I have published one or two papers in this area.

**Review Assessment: Checking Correctness Of Derivations And Theory:**

N/A

**Review Assessment: Checking Correctness Of Experiments:**

I carefully checked the experiments.

**Review Assessment: Thoroughness In Paper Reading:**

I read the paper thoroughly.

---

> ### Author Response · Authors · 2019-11-14
> **Response to Review #1**
>
> We greatly appreciate your kind and detail review. We have tried our best to meet the comments. We will respond to your concerns below in the same order they were made.
>
> 1) It actually generates relevant captions with respect to the question and answers instead of real explanations. It's true that correlated caption can sometimes serve as the explanation when they cover the same concept coincidently but, we can not use these captions to explain the reasoning process of VQA:
>
> Thank you for your kind and detailed comment.
> We stated on page 2 of the manuscript that the proposed method produces introspective explanations. However, we agree that, as the reviewer pointed out, explanations created in the proposed way are not introspective. For this reason, we have revised the content of this paper as follows: (the proposed method not only produces more introspective explanations, ... -> the proposed method not only produces more appropriate explanations, ...).
>
> 2) The technique novelty of the proposed paper is also limited; the major novelty is the VCBS, which seems very similar to CBS. The only difference is VCBS adds relaxed parameters, which seems no technique novelty.
>
> VCBS differs from CBS in that it uses loosening parameters and the number of satisfied constraints ( | y and C | ). Therefore VCBS helps to identify the keywords that cause the VQA's answer to be "no." It is the first attempt to determine the cause of "no," which is extremely hard for existing methods.
>
> We also leveraged the NLI model and ConceptNet to create explanations that require common sense. For these aforementioned reasons, we think our proposed scheme is novel enough.
>
> 3) Most annotations in Algorithm 1 is also not explained, making the readers hard to follow the actual content.
>
> Thank you for your thoughtful comment. We have revised the paper to make it easier for readers to understand the meaning of each notation in Algorithm 1.
>
> 4) The proposed model, although tied the VQA words with the explanation words, suffers the same problems as the PJ-X model, which didn't consider the VQA attention at all.
>
> Although the BUTD is used as an explanation generator, the method presented in this paper (i.e., VCBS and NLI) can also utilize the PJ-X or the Multi-VQAE as an explanation generator. To illustrate this, we conducted an additional experiment using Multi-VQAE as an explanation generator and included sample results in the Appendix.
>  As a result of replacing the explanation generator of our method to Multi-VQAE, we show that we can improve the explanations generated by the model (Multi-VQAE) while considering VQA attentions.
>
>
> 5) The experiment is also weak, considering the results is conduct on 100 samples, there might be significant variance.
>
> Thank you for your thoughtful consideration. We experimented with 100 additional samples and added the results to the Appendix. We have confirmed that we get results similar to the previous experiments.
>
> 6) It's interesting the compared approach is learned based on a different dataset, which makes the results harder to compare. The NLI model results are interesting, but for a more fair comparison, I would expect the proposed method compare with a model trained with VQA and coco caption dataset, such as VQA-E.
>
> Thank you again for your sincere advice. We understand what the reviewer is worried about. We post additional results using the Multi-VQAE as an explanation generator in the Appendix. As a result of applying our method to Multi-VQAE, we can see that more appropriate explanations are generated

---

### Decision · Program_Chairs · 2019-12-19

**Decision:**

Reject

**Comment:**

This paper is good, with relatively positive support from the reviewers. However, there were also several legitimate issues raised, for example regarding the semantics of a negative answer and associated explanations. Though this paper cannot be accepted at this time, we hope the feedback here can help improve a future version, as all reviewers agree this is a valuable line of work.